# Effects of Spatial Pattern of Forest Vegetation on Urban Cooling in a Compact Megacity

**Wen Zhou, Fuliang Cao * and Guibin Wang**

Co-Innovation Center for Sustainable Forestry in Southern China, Nanjing Forestry University;
159 Longpan Road, Nanjing 210037, China; wenzhou0305@hotmail.com (W.Z.); guibinwang99@163.com (G.W.)
* Correspondence: fuliangcaonjfu@163.com; Tel.: +86-18105170000

**Abstract:** Urban forests can be an effective contributor to mitigate the urban heat island (UHI) effect. Understanding the factors that influence the cooling intensity of forest vegetation is essential for creating a more effective urban greenspace network to better counteract the urban warming. The aim of this study was to quantify the effects of spatial patterns of forest vegetation on urban cooling, in the Shanghai metropolitan area of China, using correlation analyses and regression models. Cooling intensity values were calculated based on the land surface temperature (LST) derived from remote sensing imagery and spatial patterns of forest vegetation were quantified by eight landscape metrics, using standard and moving-window approaches. The results suggested that 90 m × 90 m was the optimal spatial scale for studying the cooling effect of forest vegetation in Shanghai's urban area. It also indicated that woodland performed better than grassland in urban cooling and the size, shape, and spatial distribution of woodland patches had significant impacts on the urban thermal environment. Specifically, the increase of size and the degree of compactness of the patch shape can effectively reduce the LST within the woodland. Areas with a higher percentage of vegetation coverage experienced a greater cooling effect. Moreover, when given a fixed amount of vegetation covers, aggregated distribution provided a stronger cooling effect than fragmented distribution and increasing overall shape complexity of woodlands can enhance the cooling effect on surrounding urban areas. This study provides insights for urban planners and landscape designers to create forest adaptive planning strategies to effectively alleviate the UHI effect.

**Keywords:** cooling effect; urban forest; landscape metrics; urban heat island

---

## 1. Introduction

Urbanization has taken place at an unprecedented rate around the world in the past few decades. More than half of the world's population now lives in towns and cities and the population will swell to about 5 billion by 2030 [1]. Much of this urbanization will take place in Africa and Asia, bringing huge social, economic, and environmental changes [2]. One of the noticeable impacts of rapid urbanization on the environment is the urban heat island (UHI) effect [3], which is a phenomenon that urban areas have higher temperatures compared to the surrounding rural areas [4–6]. Increased temperatures due to the UHI effect leads to reduced thermal comfort for urban dwellers and increased energy consumption for cooling urban indoor environments [7]. The UHI phenomenon also serves as a trap for outdoor atmospheric pollutants and affects the quality of life of urban dwellers [8,9]. Thus, UHI mitigation is urgent in order to improve the urban living environment [10].

Urban vegetation has consistently demonstrated a considerable cooling effect on mitigating urban warming, from courtyard scale to urban scale [11–16]. Many previous studies have suggested that increasing vegetation area and density is an effective strategy to reduce or alleviate the effects of urban warming [12,17–20]. Vegetated space, mainly through direct shading and evapotranspiration,

can reduce temperatures and form an urban cool island (UCI) [21–24]. Additionally, vegetated spaces transform a small part of the absorbed solar radiation by photosynthesis, instead of converting that part to heat energy. For example, Yilmaz et al. (2007) found the mean temperature of the urban forest was lower than the urban area and the temperature contrast was mainly caused by the canopy and evapotranspiration effects of the urban forest [25].

The UCI usually refers to the phenomenon that vegetated spaces have lower temperature than that of the surrounding built-up areas [26]. Generally, there are two types of UCI based on the data source applied as follows: Air/atmospheric UCI and surface UCI. The air UCI related studies use atmospheric temperature obtained from fixed weather stations or mobile equipment [27,28]. Observations based on atmospheric temperature have demonstrated that the size and shape of vegetation patches, as well as the tree species, are significant factors to influence the cooling effects [18,29,30]. Those field-based observations have found that urban greenspaces are 1 °C–7 °C cooler than the surrounding areas [31–35]. However, the acquisition of air temperature data is normally restricted by field conditions and the amount of equipment needed, which has limited the research subjects to one individual greenspace or a small number of green sites [36–39]. Therefore, quantifiable cooling effects and statistical relationships cannot be established based on these site-specific studies, which does not allow recommendations to be generated on how to best incorporate forest vegetation into an urban area for reducing temperatures [28,40].

Previous studies regarding surface UCIs mostly use land surface temperature (LST) derived from infrared remote sensing imagery. Currently, various types of satellite images and developed GIS technologies have been applied to retrieve the LST and detailed land-use and land-cover (LULC) information [41–43]. Studies of Chen et al. (2012) and Schwarz et al. (2012) have demonstrated that LST derived from remotely sensed thermal infrared data are positively and significantly correlated with field-based atmospheric temperature [44,45]. Since the LST data across the study area is time synchronized and spatially continuous and also can be easily combined with LULC map [46,47], it has been widely applied to study the relationship between landscape patterns and the LST [48]. Relating the LST data to land cover patterns has enabled the development of urban level climate adaptive strategies. In this study, we focused on remotely sensed LST and, consequently, surface UCI. Additionally, most previous studies [36,40,49–51] have focused on providing qualitative descriptions of thermal patterns or simple correlations between the LST and the land cover characteristics and only few have attempted to explore the optimal spatial patterns of urban forests to cool the surrounding urban areas [26,52,53], especially at a fixed regional scale.

The application of landscape pattern metrics greatly advanced the approach of quantifying the relationship between vegetation spatial heterogeneity and its cooling effects [54,55]. Some previous studies have suggested that the cooling effect was scale dependent [56–59]. Specifically, the term "scale" here normally refers to the spatial resolution of remotely sensed imagery and the spatial extent [60]. Several studies have indicated that remotely sensed imagery with different spatial resolutions would influence the quantification of landscape patterns [61,62]. The study of Li et al. (2013) reported that remotely sensed imagery with a higher resolution could more accurately quantify the spatial pattern of greenspace [48]. In general, previous studies have primarily focused on the impacts of spatial resolution on the statistical relationship between urban green patterns and LST, but the optimal spatial extent for its study was undefined in current literature.

As past research illustrates, the influence mechanism of spatial patterns of forest vegetation on urban cooling was not fully explored. As a result, only sporadic recommendations and strategies have been obtained to guide urban forest and land-use planning to mitigate the UHI effects. Therefore, the main objectives of this study were the following: (1) To identify the optimal spatial extent for examining the cooling effect of urban forests in Shanghai; (2) to investigate the main patch characteristics that influence the surface thermal environment within urban woodlands; (3) to explore the optimal spatial composition and configuration of woodlands that can effectively cool the urban areas.

## 2. Study Area

Shanghai, the largest city and the economic center of China, is located on the coast of the East China Sea. The Shanghai metropolitan region covers a total area of approximately 6340.5 km$^2$ and had a permanent resident population of about 24 million, as of 2015 [63]. It is largely situated on a broad flat alluvial plain with a few remnant hills in the southwest and the average elevation is about 4 m above the sea level [64]. Shanghai has a subtropical monsoon climate with a mean annual temperature of 17.1 °C and a mean annual precipitation of 1166.1 mm [65]. The regional vegetation mainly consists of evergreen broadleaved forests and evergreen broadleaved-deciduous broadleaved mixed forests [57]. Shanghai has experienced a remarkable increase in urban area over the past 50 years [66]. Currently, Shanghai is one of the most urbanized cities in China and the degree of urbanization was defined as the percentage of the total population living in urban areas [67]. Our study focused on the urbanized area within the Shanghai center city, an area of about 430 km$^2$ (Figure 1).

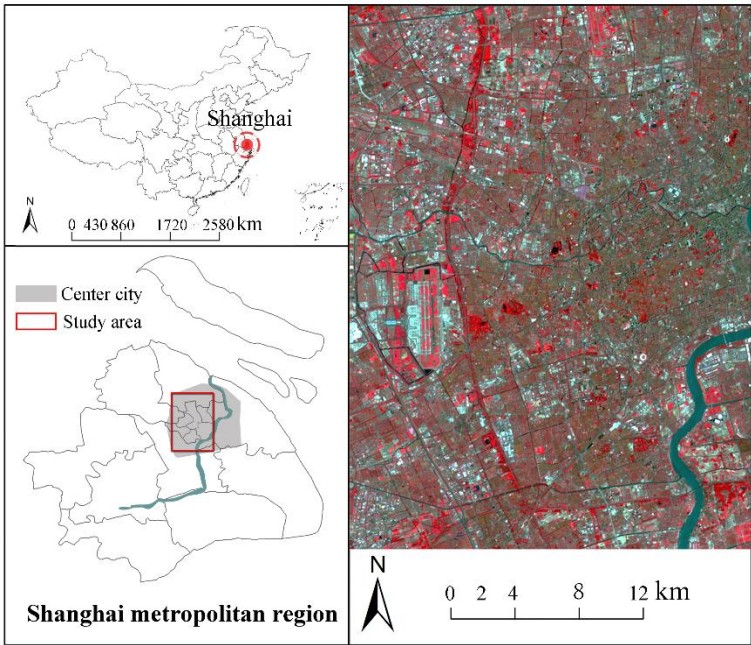

**Figure 1.** Location map of Shanghai and the study area. The false color Spot 6 image is displayed with RGB composition of band 5 (near infrared), band 4 (red), and band 3 (green).

There are several reasons of choosing this area as the study area. First, the center city (areas within the outer ring road of the city) is highly urbanized with very intensive and concentrated urban human activities. The impacts of various human activities incur many ecological and environmental problems, such as urban heat islands and water and air pollution. Therefore, the development of strategies to adapt to and mitigate the UHI effect is crucial and urgent. Second, the landscape pattern is highly heterogeneous and the distribution of vegetated areas is concentrated in some areas and scattered in others, which is favorable to study the influence of different spatial patterns of vegetated areas on LST. Last, we applied a moving-window method in this study. The moving-window method is described in Section 3.5. Therefore, the boundary effects were unavoidable. This means that only cells in which the entire (square) window is contained within the landscape are evaluated, which suggests that squared study area is the optimal choice to conduct this method.

## 3. Materials and Methods

### 3.1. Remote Sensing Data and the Pre-Processing

One Spot 6 image taken at 10:08 a.m., 25 July 2016, with four multiple spectral bands (6 m resolution) and one panchromatic band (1.5 m resolution) were used for LULC mapping, due to its high spatial resolution (1.5 m after image fusion). Additionally, a cloud-free Landsat-8 Thermal Infrared Sensor (TIRS) image (Row/Path: 038/118, 10:24 a.m., 20 July 2016) from the United States Geological Survey (https://glovis.usgs.gov/) was used for LST retrieval, since its thermal channel (30 m resolution) is widely applied to retrieve LST data. July 20th and July 25th both represented summer, the hot and humid season. Both Spot 6 images and the Landsat-8 TIRS image were clear and nearly free of clouds. Besides, the acquisition dates of the two types of remotely sensed images are only five days apart and, thus, we assumed that there was no change in the urban vegetated areas. These images were all geometrically rectified to same projection system (Datum: WGS 84 / UTM zone 51N) using ENVI software Version 5.3 (Exelis Inc.: Tysons Corner, VI, USA).

### 3.2. LULC Classification

Based on the Spot 6 fused image (combined multispectral and panchromatic bands), an urban LULC map was created by the object-oriented classification method, combined with manual visual interpretation. The study area was classified into five land-use types including the following: Woodland (trees with shrubs and grasses), grassland (shrubs and grasses), water, impervious surfaces (roads and buildings), and barren land (land without vegetation cover, mainly including exposed soil and landfill sites) (Figure 2a). Field surveys were also conducted to ascertain some doubtful pixels to improve the accuracy of the classified map. Next, 30 samples for each land cover type, 150 samples in total, were selected using a random stratified method to examine the accuracy of the classified map. Land use survey data derived from historical aerial photos, and a 1:250,000 digitalized land use map acquired in 2016 were used as the reference data. As a result, the overall accuracy of LULC classification was 94%.

### 3.3. Retrieval of LST

The Landsat 8 TIRS imagery was applied to retrieve LST. The TIRS image was rectified to a common UTM coordinate system based on the Spot 6 images. The LST was retrieved using the Mono-Window Algorithm (MWA) [68] with the following equation:

$$T_s = [a_{10}(1 - C_{10} - D_{10}) + (b_{10}(1 - C_{10} - D_{10}) + C_{10} + D_{10})T_{10} - D_{10}T_a]/C_{10}, \tag{1}$$

where $a_{10}$ and $b_{10}$ are the coefficients and constant with values of $-67.9542$ and $0.45987$, respectively (according to Qin et al. (2001) when the atmospheric temperature is in the range of 20–50 °C) [68], $T_{10}$ represents the at-sensor brightness temperature, and $T_a$ is the mean atmospheric temperature. The values $C_{10}$ and $D_{10}$ are defined, respectively, by Equations (2) and (3) as follows:

$$C_{10} = \tau_{10}\varepsilon_{10}, \tag{2}$$

$$D_{10} = (1 - \tau_{10})[1 + (1 - \varepsilon_{10})\tau_{10}], \tag{3}$$

where $\tau_{10}$ and $\varepsilon_{10}$ represent the total atmospheric transmissivity and the land surface emissivity, respectively. The LST map was then resampled to 5 m × 5 m to match the LULC map. The result of the spatial distribution of LST is shown in Figure 2b.

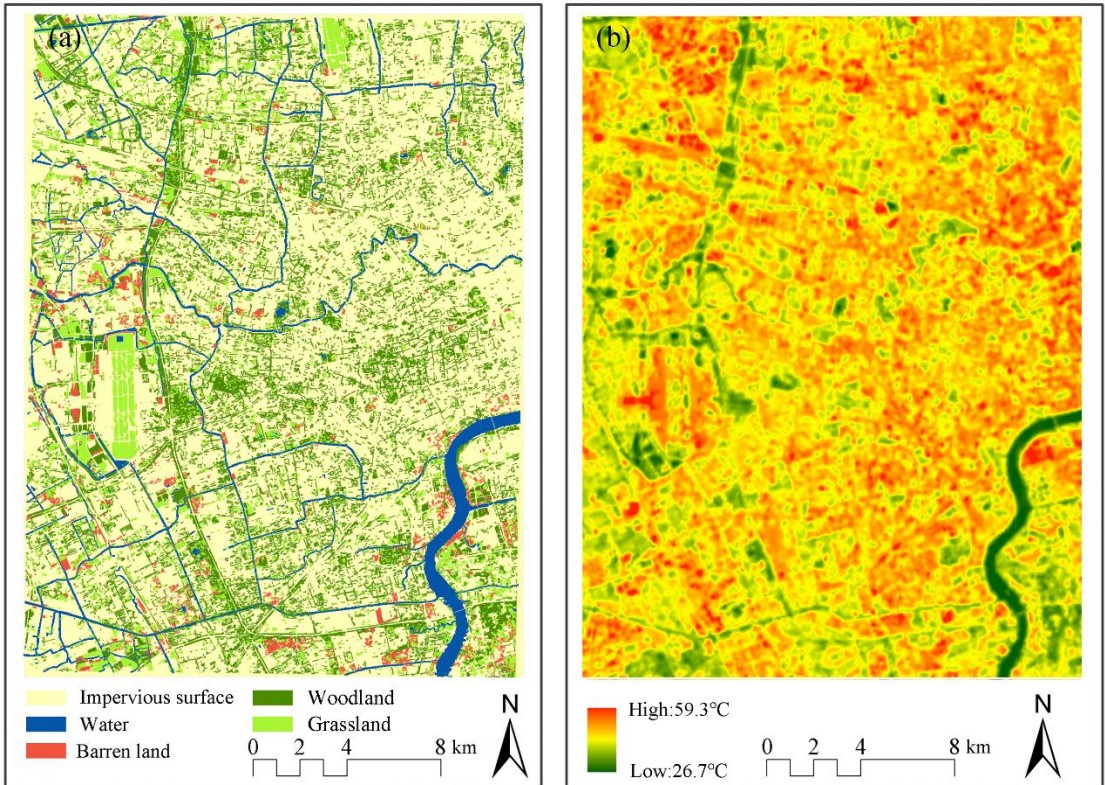

**Figure 2.** (**a**) LULC map of the study area on July 25th, 2016, based on Spot 6 images. (**b**) Spatial distribution of LST of the study area on 20 July 2016, derived from a Landsat 8 TIRS image.

### 3.4. Urban Green Pattern Metrics

The following eight landscape metrics were selected to quantify the spatial pattern of urban vegetated areas and to analyze the relationship between these patterns and corresponding LST reductions (Table 1): Patch area (PA), shape index (SI), percentage of landscape (PLAND), mean patch shape index (Shape_MN), largest patch index (LPI), mean area (Area_MN), number of patches (NP), and aggregation index (AI) [69]. PA and PLAND are composition metrics at patch level and class level, respectively. All others are spatial configuration related metrics and their corresponding application levels were shown in Table 1. The LULC map was converted to a grid format with a 5 m × 5 m cell size in order to conduct the standard and moving-window analysis in FRAGSTATS (3.3). The selection of grid size was mainly based on the resolution of Spot 6 images and, meanwhile, it has to be a divisor of the resolution of TIRS image (30 m) for later data extractions. Landscape metrics at the patch level were conducted by standard analysis in FRAGSTATS (3.3) and metrics at the class level in this study were computed by the moving-window method.

**Table 1.** Landscape metrics used to describe urban forest patterns.

| Landscape Metrics | Abbreviation | Application Levels | Description | Units |
|---|---|---|---|---|
| Patch area | PA | Patch | The area of the patch | Hectares |
| Shape index | SI | Patch | The most straightforward measure of overall shape complexity | None |
| Percentage of Landscape | PLAND | Class | The proportion of total area occupied by a particular patch type; a measure of landscape composition and dominance of patch types | Percent |
| Mean patch shape index | Shape_MN | Class | Mean value of shape index of a particular patch type | None |
| Largest patch index | LPI | Class | The area ($m^2$) of the largest patch of the corresponding patch type divided by total landscape area ($m^2$), multiplied by 100 (to convert to a percentage) | Percent |
| Mean area | Area_MN | Class | The sum of area across all patches of the corresponding patch type divided by the number of patches of the same type | Hectares |
| Number of patches | NP | Class | The number of patches of the corresponding patch type/landscape | None |
| Aggregation index | AI | Class | The number of like adjacencies involving the corresponding class, divided by the maximum possible number of like adjacencies involving the corresponding class, which is achieved when the class is maximally clumped into a single, compact patch; multiplied by 100 (to convert to a percentage) | Percent |

### 3.5. Moving-Window Analysis and Window Size Chosen

A moving window with specified shape (e.g., square in this study) and size was passed over the entire study area and values were returned to the focal cell to output a new complete and continuous grid map for each selected metric at the class [70]. Since previous studies have found that the cooling effect of green areas was scale dependent, the selection of the optimal scale of the moving window is crucial. As the resolution of TIRS band is 30 m, the tested window sizes were chosen as integer multiples of 30 m. Moreover, the mean patch size of woodland in the study area is 3800 $m^2$ (about the same size as a 60 m × 60 m window) and the data analysis showed that 91.2% of the woodland patches were smaller than 8100 $m^2$ (same size as a 90 m × 90 m window). In order to avoid dividing urban green patches into small parts by moving the window, four window sizes that were all greater than 60 m × 60 m were chosen and tested. As a result, four window sizes/spatial extents were tested (90 m × 90 m, 180 m × 180 m, 360 m × 360 m, and 720 m × 720 m) to find the optimal scale to investigate the relationship between spatial patterns of forest vegetation and temperature difference—the difference between the mean LST of each window unit and the mean LST of the study area.

### 3.6. The Calculation of Cooling Intensity

In this study, the mean LST ($T_m$) of the study area, excluding all water bodies, was treated as the reference land surface temperature. Hence, we defined the cooling intensity of woodland at patch level as $\triangle T_g = T_g - T_m$, where $T_g$ is the mean LST of a woodland patch. The mean LSTs of different land-use types were captured using the zonal analysis tool in ArcMap software. Moreover, the cooling intensity of vegetated areas to the surrounding urban area was calculated as $\triangle T = T_i - T_m$, where *Ti* is the mean LST of each window/spatial extent unit over entire study area. The mean LST of each

window patch was calculated by extracting the raster center values after resampling the LST map to corresponding window sizes.

### 3.7. Statistical Analysis

Statistical analyses were performed using the SPSS 23.0 (IBM: New York, NY, USA). Pearson correlation analysis was carried out to examine the relationship between landscape metrics and temperature reduction. Sample analyses were conducted to investigate the impact of SI on the cooling intensity. Linear regression analysis was conducted to further reveal the relationship between SI and PLAND of woodlands with temperature reduction. Window units with different values of PLAND of woodland were classified into several classes including the following: Pland_10 (9%–11%), Pland_20 (19%–21%), PA_30 (29%–31%), and Pland_40 (39%–41%). Since the study area was divided into fixed window units (e.g., 90 m × 90 m in this study), it was really hard to find sufficient sample units that contained exactly the same amounts of vegetation cover and, therefore, the scope of the PLAND value was expanded.

## 4. Results

### 4.1. The Optimal Spatial Extent for Examining the Cooling Effects of Forest Vegetation

The percentages of landscape (PLAND) of woodland were calculated by the moving-window method and showed significant linear relationships with temperature differences under four tested window sizes/spatial extents (Figure 3). The correlation coefficients of temperature differences and PLAND decreased when the window sizes increased. It may be because the PLAND values were unevenly distributed under greater spatial extents. Since it showed a good fit in Figure 3a ($R^2 = 0.501$) and also contained sufficient samples with relatively even distribution, the window size of 90 m × 90 m was decided as the optimal spatial extent in this study and was applied to subsequent analysis in Section 4.3.

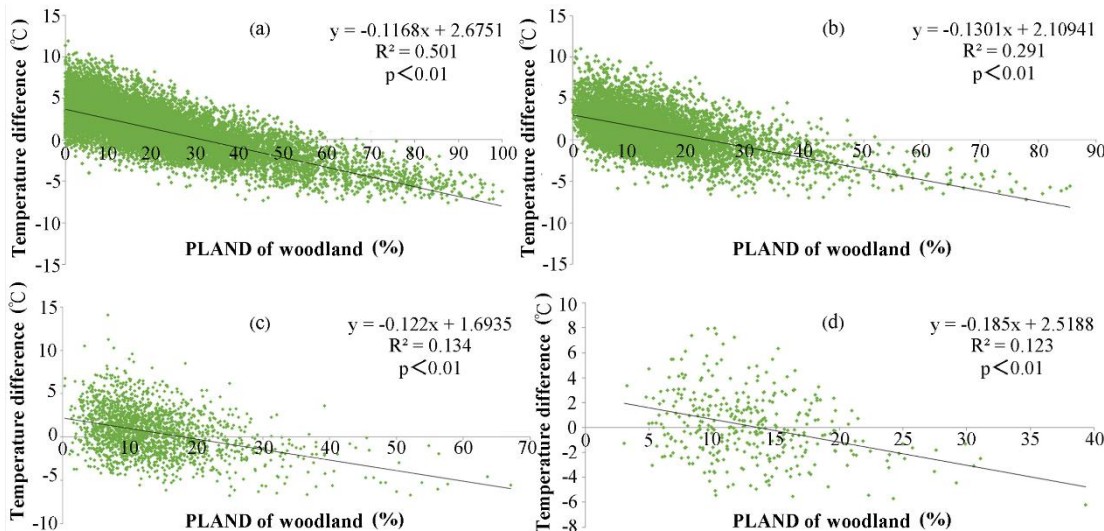

**Figure 3.** (**a**–**d**) Relationship between the percentage of landscape (PLAND) of woodland and the temperature difference (°C) in (**a**) 90 m × 90 m, (**b**) 180 m × 180 m, (**c**) 360 m × 360 m, and (**d**) 720 m × 720 m window sizes.

### 4.2. Effects of Area and Shape of Urban Woodland Patches on Cooling Intensity

The results suggested that the thermal pattern was closely related to the distribution of LULC types. Similar to previous studies [50,56], water had the greatest cooling effect (−5.23 °C) followed by woodland (−2.19 °C) and grassland (−0.34 °C) compared to the mean LST. Besides, the mean

LST of impervious surfaces was 4.87 °C higher than the mean LST, which was demonstrated as the main contributor of urban warming (Table 2). This study was targeted for urban forest vegetation and grassland showed a relative weak cooling effect and is highly attached to woodland. The following sections of this paper focus only on woodland.

**Table 2.** Results of the mean LST of different land-use types and the cooling effect (°C).

| LULC Type | Mean LST | Standard Deviation of LSTs | Temperature Reduction Compared with The mean LST of Study Area | Temperature Reduction Compared with the Mean LST of Impervious Surface |
|---|---|---|---|---|
| Woodland (*n* = 7049) | 40.23 | 2.05 | −2.19 | −7.06 |
| Grassland (*n* = 4098) | 42.08 | 1.96 | −0.34 | −5.21 |
| Impervious surface (*n* = 2786) | 47.29 | 1.52 | 4.87 | 0 |
| Barren land (*n* = 495) | 43.52 | 0.89 | 1.1 | −3.77 |
| Water (*n* = 1250) | 37.19 | 1.01 | −5.23 | −10.1 |

The temperature will represent mixed pixels when the patch size is smaller than the pixel size of the LST map (30 m × 30 m), therefore, all patches with an area smaller than 0.5 ha were excluded for analysis. As a result, 845 woodland patches were selected for sample analysis. Pearson correlation analysis was then carried out between the patch area (PA) of woodlands and their cooling intensities, and the result suggested that PA was significantly correlated with the cooling intensity ($r$ = 0.65 **, $p$ < 0.01). As shown in Figure 4, the mean LSTs of woodland patches were presented by nine PA categories, including 0.5–2 ha (*n* = 403), 2–4 ha (*n* = 282), 4–6 ha (*n* = 60), 6–8 ha (*n* = 24), 8–10 ha (*n* = 22), 10–12 ha (*n* = 18), 12–14 ha (*n* = 18), 14–16 ha(*n* = 10), and 16–18 ha (*n* = 8) in sample plot and indicated that the PA is an important factor to influence the LST. The greatest cooling intensity of woodland patch in the study area was −7.82 °C. In general, the cooling intensities of larger woodland patches were stronger and also more stable than smaller ones. In addition, the cooling intensities were varied among woodland patches with similar sizes, especially for small sizes. The results also indicated that there may exist a maximum threshold with regard to the area of woodland patches.

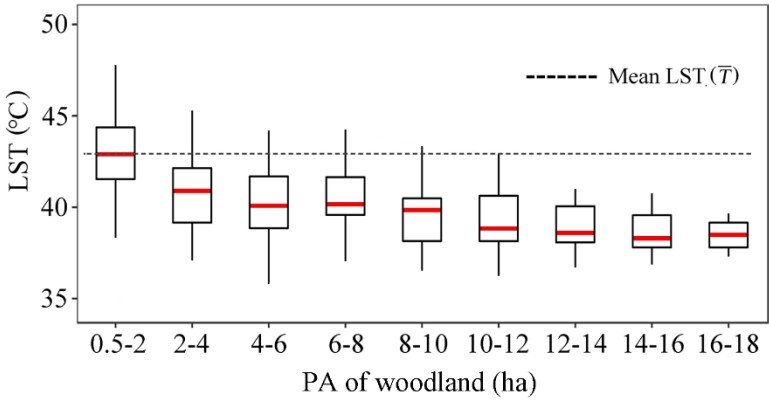

**Figure 4.** Land surface temperature (LST) by patch area (PA) of woodland in sample plot.

In order to better understand the effect of shape of woodland patches on cooling intensity by different sizes, this study classified the woodland patches into several patch area (PA) classes including the following: PA_01 (0.9–1.1 ha), PA_05 (4.9–5.1 ha), PA_10 (9–11 ha) and PA_15 (13–17 ha). The results indicated that the effect of SI on the cooling intensity were statistically significant for PA_01 ($R^2 = 0.54$, $p < 0.01$), PA_05 ($R^2 = 0.72$, $p < 0.01$), PA_10 ($R^2 = 0.63$, $p < 0.01$), and PA_15 ($R^2 = 0.53$, $p < 0.01$) and all showed significant linear relationships (Figure 5). The SI was negatively correlated with the cooling intensity.

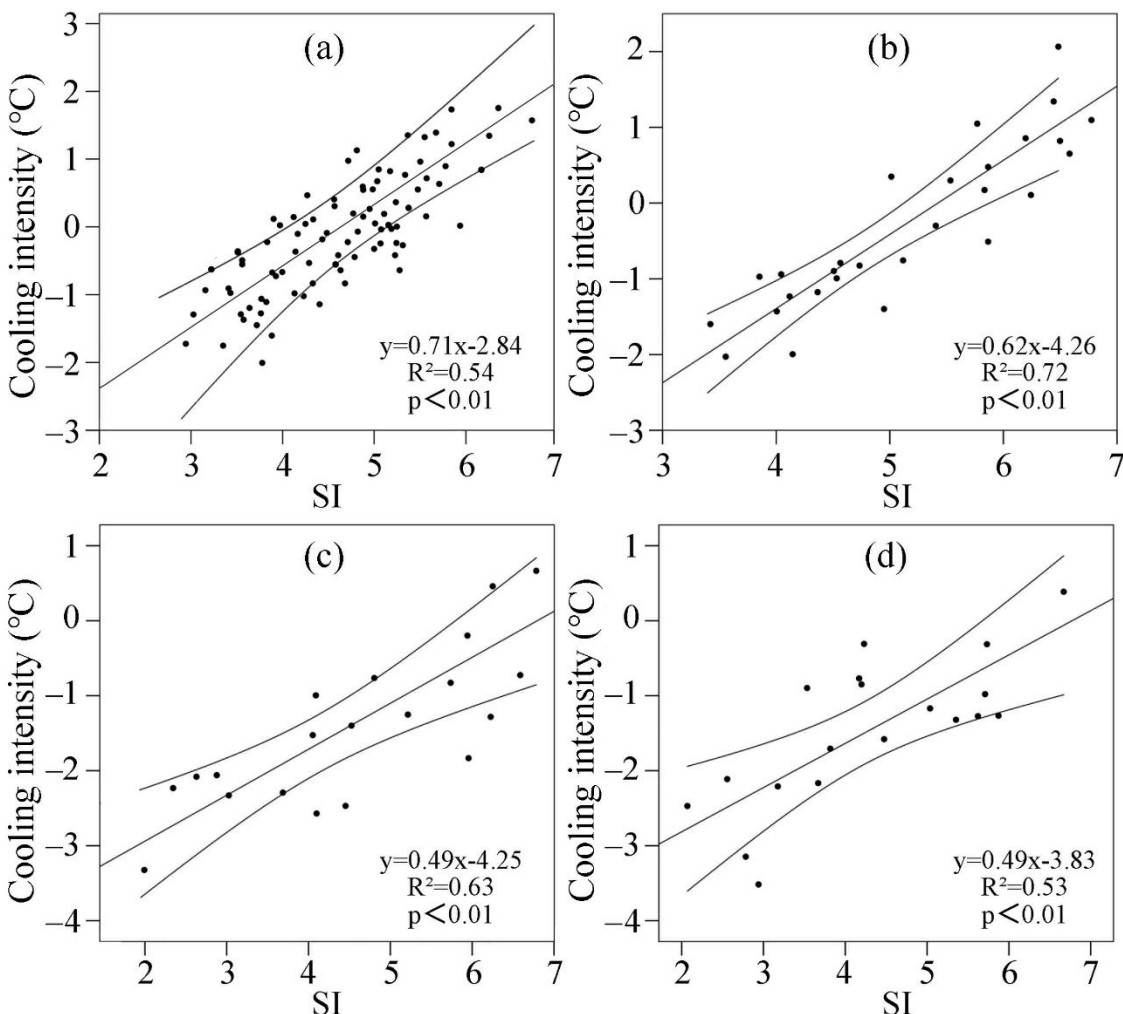

**Figure 5.** (**a**–**d**) Relationship between the cooling intensity (°C) and SI of (**a**) PA_01, (**b**) PA_05, (**c**) PA_10, (**d**) PA_15 of woodland.

### 4.3. Effects of the Spatial Pattern of Vegetated Areas on Urban Cooling

As the only composition metric in the class level, the percentage of landscape (PLAND) of woodland was confirmed to have a significant impact on urban cooling. Linear regression analysis of the relationship between PLAND and temperature reduction (Figure 3a) showed that if the PLAND of a woodland was greater than 23%, in a 90 m × 90 m spatial extent, a cooling effect will be present. Window units containing grassland and water were excluded during the sample analysis because they also provide considerable cooling effects on surrounding areas. Several pattern metrics, including Shape_MN, LPI, Area_MN, NP, and AI, were computed to explore the influence of shape, fragmentation and aggregation of woodlands on urban cooling when the amount of vegetation was relatively fixed. The results showed that cooling intensity has positive correlations with Shape MN, LPI, Area_MN, and AI, and has a negative correlation with NP, which demonstrated that the spatial

configuration of urban woodlands has a significant influence on the urban thermal environment (Table 3).

**Table 3.** Pearson correlations between configuration metrics and cooling intensity (°C).

| Pattern Metrics | Cooling Intensity | | | |
| --- | --- | --- | --- | --- |
| | Pland_10 (n = 580) | Pland_20 (n = 365) | Pland_30 (n = 260) | Pland_40 (n = 190) |
| Shape_MN | 0.38 ** | 0.23 ** | 0.45 ** | 0.39 ** |
| LPI | 0.23 ** | 0.31 ** | 0.43 ** | 0.10 * |
| Area_MN | 0.20 ** | 0.26 ** | 0.39 ** | 0.47 ** |
| NP | −0.19 ** | −0.12 * | −0.25 ** | −0.27 ** |
| AI | 0.24 ** | 0.29 ** | 0.34 ** | 0.35 ** |

\* Correlation is significant at the 0.05 level (2-tailed). \*\* Correlation is significant at the 0.01 level (2-tailed).

## 5. Discussion

Linking the method of landscape ecology and remote sensing, this study confirmed that both the composition and configuration of urban vegetated areas significantly affect the magnitude of LST in the study area. It suggested that, by balancing the amounts of vegetated areas and optimizing their spatial configuration, the cooling effect can be greatly enhanced. The main recommendations based on this study, in terms of maximizing cooling effects of greenspace, are to increase tree vegetation canopy cover and optimize green spatial patterns. Since land for urban greening is usually limited, especially for highly urbanized areas, interspersing small vegetation patches (e.g., greater than 2 ha, in this study) into urban land is also an effective method to cool the city. Apart from cooling, increasing vegetated areas can also provide some other ecological benefits, such as maintaining biodiversity, since larger greenspaces could contribute more to the conservation of biodiversity than small ones [71].

### 5.1. Implications of Optimal Spatial Extent Selection

Previous studies have confirmed that the optimal spatial scale for investigating the cooling effect of urban greenspaces is not yet known [57,59]. Since the percentage of landscape of vegetation was demonstrated as one of the key factors influencing the cooling effect [55], it was used to achieve the selection of the optimal spatial scale. In this study, a moving-window method was applied to identify the scaling effect on the relationship between urban green patterns and LSTs using landscape metrics. For this study area, a 90 m × 90 m window size was selected as the optimal spatial scale for investigating the cooling effects of forest vegetation. By studying the cooling effect under a fixed spatial scale (e.g., 90 m × 90 m square unit, in this study), it allowed specific recommendations to be generated on how to best incorporate forest vegetation in an urban area for reducing temperatures. Since no two cities are completely identical in geographical location, climate conditions, and spatial morphology, the optimal spatial scale for one city may differ from another. However, the idea and method of assessing the optimal spatial scale is replicable.

### 5.2. Implications of Patch Characteristics for Forest Adaptive Planning Strategies

Similar to the previous studies, the cooling effect of water is strongest, followed by woodland and grassland, due to lower boundary layer resistance, while impervious land is the primary contributor to the UHI phenomenon [50,56]. Results showed that the patch area and the shape index both have relationships with cooling intensity at the patch level. Specifically, the increase of area can better reduce the LST within woodland patches. The results also suggested that small woodland patches were also capable of mitigating urban warming and woodlands larger than 2 ha were normally effective in urban cooling. Additionally, the mean LSTs of smaller vegetation patches were more dissimilar than of the larger ones. The results suggested that the lower shape index of woodland can effectively enhance the cooling intensity, which means that greenspaces with relative regular and compact shapes performed

better in cooling its internal thermal environment than irregular and elongated shapes. There is no agreed statement on how the shape of urban greenspace would influence the cooling intensity. For instance, the study by Feyisa et al. (2014) suggested that regular and compact shaped parks have a stronger cooling intensity compared to irregular and elongated parks [46]. However, Chen et al. (2014) demonstrated that SI is positively correlated with temperature reduction, indicating that urban green patches with more complex shapes will perform better in urban cooling [26]. Therefore, we can reasonably infer that the selection of research areas and types of remote sensing data, as well as the acquisition time of the data, may all influence the outcomes.

### 5.3. Implications of Spatial Patterns for Forest Adaptive Planning Strategies

Previous studies have focused on how the patch characteristics of vegetated areas influence its interior thermal environment. However, fewer have investigated the effects of spatial patterns of forest vegetation on the surrounding urban areas. Based on the linear regression analysis, increasing the percentage of landscape of woodlands can effectively enhance the cooling effect. Specifically, an occupancy of woodland greater than 23%, within a 90 m × 90 m, area could create a cool island capable of counteracting the UHI effect. The results suggested that increasing the amount of vegetation cover was more effective in urban cooling than optimizing the spatial configuration, which is consistent with the findings of Zhou et al. (2011) [55]. However, the spatial configuration of woodlands also has significant impact on the magnitude of LST. When given a fixed amount of vegetation, aggregated and concentrated distribution across the urban landscape is more effective to reduce the LST than scattered and fragmented distribution. The results also suggested that a vegetation patch with an irregular and elongated shape performed better in cooling its surrounding urban areas. This is probably because when a patch area is fixed, an irregular vegetation patch is in contact with larger area of surrounding landscape, compared to a compact one. In addition, a mainland-island spatial configuration of a woodland within a 90 m × 90 m area can effectively enhance the cooling effect.

## 6. Conclusions

Most of the previous studies regarding the cooling effect of urban greenspaces focused either on a small amount of green sites or certain types of greenspace, such as urban parks [35,51], green roofs [72], and roadside greenspaces [36]. However, fewer have attempted to assess the effects of spatial patterns of forest vegetation on urban cooling, especially at urban regional scale. Hence, conclusions drawn from an individual greenspace or few examples of green sites cannot be simply transferred and applied to make planning strategies to counteract the UHI effect at an urban and larger scale. Understanding the influence mechanism of forest vegetation on urban cooling is difficult because, not only patch characteristics (e.g., size and shape), but also the spatial configuration of urban greenspaces would affect the cooling intensity, as well as the characteristics of the surrounding urban area and climate conditions [26,51,69,73]. In this study, we have applied eight landscape metrics at patch and class levels and identified the main factors that influence the cooling intensity of urban forests. By generating a more comprehensive understanding in how forest vegetation can better cool its interior thermal environment and the surrounding urban areas, it may be possible to maximize the cooling effect in urban planning from the incipient planning stage.

With rapid urbanization, sprawl invades urban vegetated areas and they are largely replaced by extensive impervious low albedo paving and building materials and, to a large extent, causes the UHI phenomenon. For those highly urbanized areas, the removal of paved surfaces and buildings are expensive and impractical [74], however, the amount and spatial configuration of vegetated areas can be altered and optimized through vegetation management, such as increasing the tree vegetation canopy cover, the greening of roofs, adding vegetation cover on parking lots, and even building walls. Strategies relating to increasing the overall patch shape complexity are effective for cooling urban environments. Moreover, optimizing the structure of vegetation planting within a vegetated space, such as increasing the compactness of vegetation communities, can effectively reduce the internal

temperature. Further quantifiable research is needed to generate compromised recommendations regarding to internal and external cooling effect of urban forests. Therefore, understanding the effects of composition and spatial configuration of forest vegetation on urban cooling is becoming increasingly important for tackling the UHI effect, particularly for areas where urbanization is still in process. The results of this study provide urban planners and natural resource managers with some theoretical and practical information on how urban forest vegetation should be planned and managed to better contradict the UHI effect.

**Author Contributions:** Conceptualization, W.Z. and F.C.; methodology, W.Z. and G.W.; software, W.Z.; validation, W.Z. and F.C.; formal analysis, W.Z.; investigation, W.Z.; resources, W.Z. and G.W.; data curation, W.Z.; writing—original draft preparation, W.Z.; writing—review and editing, W.Z., F.C and G.W.; visualization, W.Z.; supervision, F.C and G.W; project administration, F.C.; funding acquisition, W.Z. and F.C.

**Funding:** This research was funded by the National Key Research and Development Program of China (2017YFD0600701) and the Doctorate Fellowship Foundation of Nanjing Forestry University.

**Conflicts of Interest:** The authors declare no conflict of interest.

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
