# Peer review of "Effects of Spatial Pattern of Forest Vegetation on Urban Cooling in a Compact Megacity"

_forests, doi:10.3390/f10030282_

Round 1

Reviewer 1 Report

Thank you for your revised manuscript. I appreciate your efforts to improve the manuscript and to address my comments so thoroughly. There are still a few places, however, where points have not been addressed in the main manuscript, and where the new text is ambiguous.

Major points

Line 106 – urbanisation – in the Response document you provide useful clarification of how urbanisation was defined here (comment Lines 99 and 100) and why you selected your study area and its boundaries (comment line 100). This hasn’t been translated into the main text very effectively – some concise definition in the main text would be useful for anyone wanting to following equivalent methods.

Paragraph starting line 114 – The additional explanation about the satellite data processing is appreciated here. It would be useful to explain briefly why the two data sets were collected, as this is still not entirely clear – if I understand correctly, the Spot data were used for the LULC and Landsat for LST? In addition, it would be useful to briefly explain why those dates were selected – do these correspond to particularly high temperatures?

Line 127 – The additional explanation here is welcome. It is still unclear, however, how the accuracy was checked. Are these 150 locations where the field surveys were done (if so the paragraph needs restructuring)?

There are still quite a few sections in the Results that would be better in the Methods, or that need to be explained in the Methods

-          Comment (e) in the Response document – some brief version of this, about how PLAND categories were determined – would be useful in the Methods

-          A Statistical Analysis section still needs to be added to the Methods and to include a more detailed explanation of the procedures mentioned in line 211 (Pearson correlation), line 227 (sample analyses), line 237 (linear regression). What were the details of these analyses? What programs and packages were used?

Table 2 – I would still like to see some information about the spread of values, as well as the mean

The text in lines 305-308 about greenspace shape in relation to internal cooling (“The results also suggested that vegetation patch with irregular and elongated shape performed better in cooling its surrounding urban areas. This is probably because when patch area is fixed, irregular vegetation patch is in contact with larger area of surrounding landscape compared to a compact one.”) and in lines 286 to 288 about greenspace shape for cooling the surrounding area (“greenspace with relative regular and compact shape performed better in cooling its inside thermal environment than irregular and elongated shape”) lead to rather contradictory planning recommendations it seems to me. You mention, for example, the need to ‘optimize greenspace size and shape’ in the new text in line 315, but this is difficult if results are different for different aims. It would be very useful to see some discussion of this point given the focus of the manuscript on providing recommendations. It would be useful to consider adding subheadings to the Discussion as well to clearly show how each how each aim was met and any then any synthetic conclusions.

Lines 325 to 328 – this new text seems contradictory – are you saying that it is possible to plant trees despite the expense and difficulty? Or are you suggesting that changing the spatial configuration of vegetated areas is not expensive or impractical? Changing the configuration of greenspace necessarily means reducing or altering the area of buildings or paved areas. Is the point that this may be difficult but that the difficulty is worth it for the reduction in temperature and the associated benefits of greenspace?

Minor points

Line 37 – suggest changing ‘have’ to ‘leads to’

Line 38-9 – sentence starting ‘Besides’ – rephrase, this is unclear. Do you mean ‘The UHI phenomenon also…’?

Line 46 – should ‘was’ be ‘is’?

Lines 45 to 46 – sentence starting ‘Besides’ – this doesn’t appear to be a complete sentence.

Line 49 – add ‘the’ before urban forest.

Line 54 – vegetation patches

Line 55 – tree species ‘composition’?

Line 82 – there appears to be a track change remaining

Line 104 – what is ‘enormous’ urbanisation?

Line 119 – days, not days’

Line 126 – as in the examples from the Introduction – sentences starting with ‘Besides’ – do you mean ‘also’ or ‘in addition’? For example, could lines 126 be ‘Field survey were also conducted to…’

Sentence starting line 129 – this should come earlier in the paragraph.

Line 161 – check capitalisation and spaces

Line 202 – Water, change to water

Abbreviations in the Discussion: shape index (SI) is introduced twice in the Discussion (lines 281 & 285) but never used as an abbreviation, and PA and PLAND are also introduced once but the abbreviations don’t seem to be used. Probably easiest to drop these.

Line 268 – rephrase ‘have be’ and ‘as an crucial’

Lines 273-274 – re-phrase to aid comprehension.

Lines 27 and 274 - ‘Scientific’ planning – I think this term should be dropped as I don’t think it is widely understood.

Line 322 – This may be a more up-to-date reference to consult here: Beninde et al (2015). Biodiversity in cities needs space: a meta‐analysis of factors determining intra‐urban biodiversity variation. Ecology letters, 18(6), 581-592.

Author Response

1.     Reviewer: Line 106 – urbanisation – in the Response document you provide useful clarification of how urbanisation was defined here (comment Lines 99 and 100) and why you selected your study area and its boundaries (comment line 100). This hasn’t been translated into the main text very effectively – some concise definition in the main text would be useful for anyone wanting to following equivalent methods.

Author: Thanks a lot for your suggestion. We followed your suggestion and added required information about the measure of ‘urbanization’ and the reason of selecting the studied area which can be found in page 3, line 103-104, “…and the degree of urbanization was defined as the percentage of the total population living in urban areas”, and in page 3-4, line 109-120, “There are several reasons of choosing this area as the study area. First, the center city (areas within the outer ring road of the city) is high urbanized and with very intensive and concentrated urban human activities. The impacts of various human activities incur many ecological and environmental problems, such as urban heat islands, water and air pollution. Therefore, the development of strategies to adapt to and mitigate the UHI effect is crucial and urgent. Second, the landscape pattern is highly heterogeneous, and the distribution of vegetated areas is concentrated in some areas and scattered in others which is favorable to study the influence of different spatial patterns of vegetated areas on LST. Last, we applied moving-window method in this study. The moving-window method are described in Section 3.5. Therefore, the Boundary effects was unavoidable. It means that only cells in which the entire (square) window is contained within the landscape are evaluated, it suggests that squared study area is the optimal choice to conduct this method.”

2.     Reviewer: Paragraph starting line 114 – The additional explanation about the satellite data processing is appreciated here. It would be useful to explain briefly why the two data sets were collected, as this is still not entirely clear – if I understand correctly, the Spot data were used for the LULC and Landsat for LST? In addition, it would be useful to briefly explain why those dates were selected – do these correspond to particularly high temperatures?

Authors: Thank you very much for your suggestion. Yes, the Spot 6 data was used for the LULC classification and the Landsat 8 TIRS imagery was used for retrieving the LST. We followed your suggestion and provided brief explanations of reasons to choose the two types of data and their dates, and those changes can be found in page 4, line 123-129, “One Spot 6 image taken at 10:08 a.m., July 25th, 2016, with four multiple spectral bands (6 m resolution) and one panchronmatic band (1.5 m resolution) were used for LULC mapping due to its high spatial resolution (1.5 m after image fusion). Besides, a cloud-free Landsat-8 Thermal Infrared Sensor (TIRS) image (Row/Path: 038/118, 10:24 a.m., July 20th, 2016) from the United States Geological Survey (https://glovis.usgs.gov/) was used for LST retrieval since its thermal channel (30m resolution) is widely applied to retrieve LST data. July 20th  and July 25th both represented summer, the hot and humid season.”

3.     Reviewer: Line 127 – The additional explanation here is welcome. It is still unclear, however, how the accuracy was checked. Are these 150 locations where the field surveys were done (if so the paragraph needs restructuring)?

Author: We really appreciate your valuable suggestions. Before the accuracy assessment, we did massive edits by visual interpretation and “field surveys were also conducted to ascertain some doubtful pixels to improve the accuracy of classified map” (see page 4, line 140-141). We followed your suggestion and added more details on how we conduct the accuracy assessment which can be seen in page 4, line 142-144, “Land use survey data derived from historical aerial photos, and a 1:250,000 digitalized land use map acquired in 2016 were used as the reference data.”

4.     Reviewer: There are still quite a few sections in the Results that would be better in the Methods, or that need to be explained in the Methods

-Comment (e) in the Response document – some brief version of this, about how PLAND categories were determined – would be useful in the Methods

-A Statistical Analysis section still needs to be added to the Methods and to include a more detailed explanation of the procedures mentioned in line 211 (Pearson correlation), line 227 (sample analyses), line 237 (linear regression). 。。。。?

Authors: Thanks a lot for your suggestion. We followed your suggestion and removed some contents from Results section to Methods section which can be seen in the new added section: 3.7 Statistical analysis which can be seen in page 7, line 199-208, “Statistical analyses were performed using the SPSS 23.0. Pearson correlation analysis were carried out to examine the relationship between landscape metrics and temperature reduction. Sample analyses were conducted to investigate the impact of SI on the cooling intensity. Linear regression analysis was conducted to further reveal the relationship between SI and PLAND of woodland with temperature reduction. Window units with different values of PLAND of woodland were classified into several classes including: Pland_10 (9 ~ 11 %), Pland_20 (19 ~ 21 %), PA_30 (29 ~ 31 %), and Pland_40 (39 ~ 41 %). Since the study area was divided into fixed window units (e.g. 90 m x 90 m in this study), it was really hard to find sufficient sample units that contained exactly same amounts of vegetation cover, and therefore, the scope of PLAND value was expanded.”

5.     Reviewer: Table 2 – I would still like to see some information about the spread of values, as well as the mean

Author: We really appreciate your valuable suggestions. We followed your suggestion and added the number of samples involved in data analysis for each land-use type and also the standard deviation of temperature values for different land use types which can be seen in Table 2 in page 8.

6.     Reviewer: The text in lines 305-308 about greenspace shape in relation to external cooling (“The results also suggested that vegetation patch with irregular and elongated shape performed better in cooling its surrounding urban areas. This is probably because when patch area is fixed, irregular vegetation patch is in contact with larger area of surrounding landscape compared to a compact one.”) and in lines 286 to 288 about greenspace shape for cooling the surrounding area (“greenspace with relative regular and compact shape performed better in cooling its inside thermal environment than irregular and elongated shape”) lead to rather contradictory planning recommendations it seems to me. You mention, for example, the need to ‘optimize greenspace size and shape’ in the new text in line 315, but this is difficult if results are different for different aims. It would be very useful to see some discussion of this point given the focus of the manuscript on providing recommendations. It would be useful to consider adding subheadings to the Discussion as well to clearly show how each how each aim was met and any then any synthetic conclusions.

Authors: Thank you very much for your suggestion. We followed your suggestion and divided the Discussion and Conclusion section into two sections: 5. Discussion and 6. Conclusion, and added subheadings to the Discussion section including: 5.1 Implications of optimal spatial extent selection; 5.2 Implications of patch characteristics for forest adaptive planning strategies; and 5.3 Implications of spatial patterns for forest adaptive planning strategies. Besides, we readjusted the order of two paragraphs (Paragraph starting line 272 and paragraph starting line 332). We added more discussion about the recommendations of greenspace shape since the results of the internal cooling and external cooling were contradictory, and changes can be seen in page 13, line 350-354, “Strategies relating to increasing the overall patch shape complexity is effective to cool urban environments. Moreover, optimizing the structure of vegetation planting within a vegetated space such as increasing the compactness of vegetation communities can effectively reduce the internal temperature. Further quantifiable research is needed to generate compromised recommendations regarding to internal and external cooling effect of urban forests.”

7.     Reviewer: Lines 325 to 328 – this new text seems contradictory – are you saying that it is possible to plant trees despite the expense and difficulty? Or are you suggesting that changing the spatial configuration of vegetated areas is not expensive or impractical? Changing the configuration of greenspace necessarily means reducing or altering the area of buildings or paved areas. Is the point that this may be difficult but that the difficulty is worth it for the reduction in temperature and the associated benefits of greenspace?

Authors: Thank you very much for your question. We realized our expression is unclear and may lead to misunderstanding and therefore, we rewrote this part which can be seen in page 3, line 346-354, “For those highly urbanized areas, the removal of paved surfaces and buildings are expensive and impractical [74], however, the amount and spatial configuration of vegetated areas can be altered and optimized through vegetation management such as increasing tree vegetation canopy cover and the greening of roofs, and adding vegetation cover on parking lots and even building walls.”

Minor points

8.     Reviewer: Line 37 – suggest changing ‘have’ to ‘leads to’

Authors: “…have…” has been changed to “…leads to…”. (See Page 1, line 37)

9.     Reviewer: Line 38-9 – sentence starting ‘Besides’ – rephrase, this is unclear. Do you mean ‘The UHI phenomenon also…’?

Authors: “…Besides, the UHI phenomenon serves as a trap…” has been changed to “…The UHI phenomenon also serves as a trap…”. (See Page 1, line 38-9)

10.  Reviewer: Line 46 – should ‘was’ be ‘is’?

Authors: Thanks a lot for your suggestion. This sentence was rewritten as “Besides, vegetated spaces transform a small part of the absorbed solar radiation by photosynthesis instead of converting that part to heat energy”. (See Page 2, line 45-47)

11.  Reviewer: Lines 45 to 46 – sentence starting ‘Besides’ – this doesn’t appear to be a complete sentence.

Authors: Thank you very much for your reminder. We rewrote this sentence as “Besides, vegetated spaces transform a small part of the absorbed solar radiation by photosynthesis instead of converting that part to heat energy.” (See Page 2, line 45-7)

12.  Reviewer: Line 49 – add ‘the’ before urban forest.

Authors: “…urban forest…” has been changed to “…the urban forest…”. (See Page 2, line 49)

13.   Reviewer: Line 54 – vegetation patches

Authors: “…vegetation patch…” has been changed to “…vegetation patches…”. (See Page 2, line 54)

14.   Reviewer: Line 55 – tree species ‘composition’?

Authors: Thanks a lot for your question. “Observations based on atmospheric temperature have demonstrated that the size and shape of vegetation patches, as well as tree species are significant factors to influence the cooling effects” (See Page 2, line 53-55). For example, Feyisa et al. (2014) indicated that Eucalyptus sp. had a significantly higher cooling effect than any other species group (P < 0.05) and the species with the least effect on temperature were Grevillea and Cupressus. Moreover, the study of Rahman et al. (2014) demonstrated that the cooling effect of P. calleryana and C. laevigatawere were stronger than Prunus Umineko and S. arnoldiana.

Feyisa, G.L.; Dons, K.; Meilby, H. Efficiency of parks in mitigating urban heat island effect: An example from Addis Ababa. Landsc. Urban Plan. 2014, 123, 87–95.

Rahman, M.A.; Armson, D.; Ennos, A.R. A comparison of the growth and cooling effectiveness of five commonly planted urban tree species. Urban Ecosyst. 2015.

15.   Reviewer: Line 82 – there appears to be a track change remaining

Authors: Thanks a lot for your reminder. We deleted the underline mark ‘ (See Page 2, line 82)

16.  Reviewer: Line 104 – what is ‘enormous’ urbanisation?

Authors: Thanks a lot for your question. We realized the expression ‘enormous urbanization’ is inappropriate and this sentence has been changed to “Shanghai has experienced a remarkable increase in urban area over the past 50 years”. (See Page 3, line 102)

17.  Reviewer: Line 119 – days, not days’

Authors: “…days’…” has been changed to “…days…”. (See Page 4, line 130)

18.   Reviewer: Line 126 – as in the examples from the Introduction – sentences starting with ‘Besides’ – do you mean ‘also’ or ‘in addition’? For example, could lines 126 be ‘Field survey were also conducted to…’

Authors: “Besides, field surveys were conducted to…” has been changed to “…Field surveys were also conducted to…”. (See Page 4, line 140)

19.  Reviewer: Sentence starting line 129 – this should come earlier in the paragraph.

Authors: Thanks a lot for your suggestion. We followed your suggestion and moved this sentence to page 4, line 137-140, “. The study area was classified into five land-use types including: woodland (trees with shrubs and grasses), grassland (shrubs and grasses), water, impervious surface (roads and buildings) and barren land (land without vegetation cover, mainly including exposed soil and landfill sites) (Figure 2a)”.

20.   Reviewer: Line 161 (now line 174) – check capitalisation and spaces

Authors: Thank you very much for your reminder. “Moving-window Analysis” has been changed to “Moving-window analysis” (See Page 6, line 174).

21.   Reviewer: Line 202 – Water, change to water

Authors: “…Water…” has been changed to “…water…”. (See Page 7, line 224)

22.   Reviewer: Abbreviations in the Discussion: shape index (SI) is introduced twice in the Discussion (lines 281 & 285) but never used as an abbreviation, and PA and PLAND are also introduced once but the abbreviations don’t seem to be used. Probably easiest to drop these.

Authors: Thanks a lot for your suggestions. We followed your suggestions and drop these abbreviations and changes can be seen in page 11, line 283, “…the percentage of landscape…” , and line 297 “…patch area and shape index…”, and line 302 “…shape index…”, and page 12, line 316, “…the percentage of landscape…”.

23.  Reviewer: Line 268 – rephrase ‘have be’ and ‘as an crucial’

Authors: “…have be confirmed as an crucial factor regarding to UHI mitigation,…” has been changed to “…was demonstrated as one of the key factors influencing the cooling effect,…”. (See Page 10, line 284)

24.  Reviewer: Lines 273-274 – re-phrase to aid comprehension.

Authors: We rewrote this sentence as “By studying the cooling effect under a fixed spatial scale (e.g. 90 m × 90 m square unit in this study), it allowed specific recommendations to be generated on how to best incorporate forest vegetation in an urban area for reducing temperatures”. (See Page 10, line 288-290)

25.   Reviewer: Lines 27 and 274 - ‘Scientific’ planning – I think this term should be dropped as I don’t think it is widely understood.

Authors: Thanks a lot for your suggestion. We followed your suggestion and dropped the term - ‘Scientific’ planning, and changes can be seen in page 1, line 27 “…create forest adaptive planning strategies”, and in page 10, line 289-290 “…it allowed specific recommendations to be generated on how to best incorporate forest vegetation in an urban area for reducing temperatures”.

26.   Reviewer: Line 322 – This may be a more up-to-date reference to consult here: Beninde et al (2015). Biodiversity in cities needs space: a metaanalysis of factors determining intraurban biodiversity variation. Ecology letters, 18(6), 581-592.

Authors: Thank you very much for your suggestion. We followed your suggestion and updated the reference NO. 71 which can be seen in page 11, line 280.

Reviewer 2 Report

Overall the authors made substantive effort to improve the manuscript and almost ready now to be accepted. I felt the some of the comments (from previous version) were not dealt, for instance:

Line 19: What do you mean by “spatial patterns” here? The effect size?

Line 21: What is decrease of shape? Is it for instance from triangular to liner or something like this?

Author Response

1.     Reviewer: Line 19: What do you mean by “spatial patterns” here? The effect size?

Author: Thanks a lot for your question. “Spatial patterns of woodland” means the size, shape and spatial distribution of woodland patches. Therefore, we changed “…spatial patterns of woodland…” to “…the size, shape and spatial distribution of woodland patches…”. (See Page 1, line 19-20)

2.     Reviewer: Line 21: What is decrease of shape? Is it for instance from triangular to liner or something like this?

Author: Thank you very much for your question. In our study, we used shape index (SI) to describe the shape complexity of each vegetation patch, and the corresponding description of SI can be found in Table 1 (page 6). SI = 1 when the patch is maximally compact as a circle and SI = 1.13 as a square, and SI increases without limit as patch shape becomes more irregular as shown in diagram below, therefore, “…decrease of shape complexity” means a patch becomes more regular. We realized our expression here may not easy to be understood, therefore we changed “…the increase of size and decrease of shape complexity…” to “…the increase of size and the compactness degree of the patch shape…” (see page 1, line 21).